# Reduced SIRT1 and SIRT3 and Lower Antioxidant Capacity of Seminal Plasma Is Associated with Shorter Sperm Telomere Length in Oligospermic Men

**DOI:** 10.3390/ijms25020718

**Published:** 2024-01-05

**Authors:** Varinderpal S. Dhillon, Mohammad Shahid, Permal Deo, Michael Fenech

**Affiliations:** 1Health and Biomedical Innovation, UniSA Clinical and Health Sciences, University of South Australia, Adelaide 5000, Australia; permal.deo@unisa.edu.au (P.D.); michael.fenech@unisa.edu.au (M.F.); 2Department of Basic Medical Sciences, College of Medicine, Prince Sattam bin Abdulaziz University, Al-Kharj 11942, Saudi Arabia; dr.shahid90@yahoo.com

**Keywords:** oxidative stress, seminal plasma, sperms, telomere length, antioxidant capacity

## Abstract

Infertility affects millions of couples worldwide and has a profound impact not only on their families, but also on communities. Telomere attrition has been associated with infertility, DNA damage and fragmentation. Oxidative stress has been shown to affect sperm DNA integrity and telomere length. Sirtuins such as SIRT1 and SIRT3 are involved in aging and oxidative stress response. The aim of the present study is to determine the role of SIRT1 and SIRT3 in regulating oxidative stress, telomere shortening, and their association with oligospermia. Therefore, we assessed the protein levels of SIRT1 and SIRT3, total antioxidant capacity (TAC), superoxide dismutase (SOD), malondialdehyde (MDA) and catalase activity (CAT) in the seminal plasma of 272 patients with oligospermia and 251 fertile men. We also measured sperm telomere length (STL) and leukocyte telomere length (LTL) using a standard real-time quantitative PCR assay. Sperm chromatin and protamine deficiency were also measured as per standard methods. Our results for oligospermic patients demonstrate significant reductions in semen parameters, shorter STL and LTL, lower levels of SOD, TAC, CAT, SIRT1 and SIRT3 levels, and also significant protamine deficiency and higher levels of MDA and DNA fragmentation. We conclude that a shorter TL in sperms and leukocytes is associated with increased oxidative stress that also accounts for high levels of DNA fragmentation in sperms. Our results support the hypothesis that various sperm parameters in the state of oligospermia are associated with or caused by reduced levels of SIRT1 and SIRT3 proteins.

## 1. Introduction

Infertility is a major health issue worldwide, affecting approximately 8–12% of couples in the reproductive age group who are unable to conceive at least once after 12 months or more of regular unprotected sexual intercourse [1,2]. Overall, infertility affects up to 80 million people worldwide, with 50% being due to male-infertility-associated factors related to abnormal semen parameters [3,4]. Sperm count, percentage of motile sperm and abnormal sperm morphology in the ejaculate are used to determine semen quality [3]. Oligoasthenoteratozoospermia (OAT), i.e., low sperm count is a common type of disease in patients with male infertility, and approximately 3/4 of infertile men seeking help have idiopathic OAT [5].

Accelerated ageing and/or high exposure to endogenous or exogenous genotoxins may be major causes of infertility [6,7]. Sirtuins may prevent hallmarks of aging such as DNA damage, genomic instability, metabolic syndrome, neurodegenerative diseases, chronic inflammatory response and cancer [8,9,10,11]. In mammals, there are seven sirtuins (SIRT1–SIRT7) that have different sub-cellular locations, chemical structures, or target proteins [12]. SIRT1 and SIRT2 are located in the cytoplasm and nucleus, and SIRT6 and SIRT7 exist in the nucleus only, whereas SIRT3-SIRT5 are located in the mitochondria [13]. Sirtuins, particularly SIRT1 and SIRT3, are highly expressed in mammalian testicular tissue, probably because spermatogenesis depends on glycolysis and the generation of ATP by pyruvate metabolism in mitochondria [14,15,16]. Immuno-blotting positive staining in late spermatogonia, spermatocytes, round spermatids and pachytene spermatocytes indicates that SIRT1 is a nuclear protein, and it plays an important role in the development of germ cells during spermatogenesis [17,18]. A gene ontology study in *Sirt1*^−^*^/*−*^* mice has provided further evidence that many genes involved in spermatogenesis are differentially expressed in testis, leading to the suggestion that SIRT1 deficiency influences fertility by regulating the transcription of several spermatogenic genes [15,17]. SIRT1-mediated deacetylation can influence sumoylation, which regulates many cellular physiological activities in testis and spermatogenesis, such as germ cell proliferation, heterochromatin remodeling, and changes in nuclear morphology [19,20,21,22,23]. SIRT1 as well as SIRT3 and peroxisome proliferator-activated receptor γ coactivator 1 alpha (PGC1α) activate antioxidant defense systems; therefore, it can be assumed that abnormal spermatogenesis in *Sirt1^−/−^* mice might be due to oxidative stress [24]. For normal spermatogenesis, both glucose metabolism and lactate synthesis must be optimal; however, excessive glycolytic activity may lead to mitochondrial ROS overproduction [25,26]. Moreover, the loss of SIRT1 and/or SIRT3 causes a malfunctioning electron transport chain, and a reduction in the activity of antioxidant defenses [16]. Sperm membranes contain a large proportion of polyunsaturated fatty acids (PUFA), and hence are extremely vulnerable to oxidative stress [27]. Optimal fluidity and fusogenicity are required for acrosomal processes, and sperm–oolemma interactions are controlled by PUFAs that are highly susceptible to lipid peroxidation [28,29]. It has been shown that decreased SIRT1 expression induces defects in acrosome formation leading to abnormal sperm morphology [30]. Recently, it has been shown that SIRT1 and SIRT3 proteins are inversely correlated with oxidative stress biomarkers and sperm DNA fragmentation in infertile patients [31,32]. Recent evidence suggests that telomere erosion depresses sirtuin activity. Conversely, increasing sirtuin activity stabilizes telomeres and improves telomerase-dependent disease states [33] (Amano and Sahin, 2019). Sirtuins (SIRT1) is mobilized from telomeres to sites of DNA double strand breaks. It has been shown that SIRT1 inhibits telomerase activity, which would be expected to drive the cell to senescence or apoptosis. SIRT1 depletion also results in increased genome instability and telomeric aberrations that contribute to decreased cell growth [34]. It has been shown that sirtuins modify histones, epigenetic enzymes, transcription factors, and transcription factor regulators to control gene expression [35]. Sirtuins deacetylate transcription factors to control patterns of gene expression by deacetylating the lysine residues present on transcription factors such as p53, FoxO1, FoxO3, FoxO4, FoxO6, and NF-κB [36].

Reactive oxygen species (ROS) in sperms originate from activated leukocytes present in the seminal plasma, and from the mitochondria [37]. Sperms may be highly susceptible to damage caused by high levels of oxidative stress, mainly due to the low levels of antioxidants defense of sperms, and evidence suggests that ROS-mediated damage is the main cause of dysfunctional sperm in the majority of infertile patients [38].

Telomeres are repetitive hexameric DNA sequences (TTAGGG)n at the end of chromosomes and play an important role in maintaining genomic stability by enabling T-loop formation and the assembly of the shelterin complex that protects against telomere attrition and malfunction [39,40]. Excessive telomere shortening or loss of its functionality is associated with chromosome end-to-end fusions, resulting in dicentric chromosome formation, which causes mitotic disruption via anaphase bridge formation resulting in increased chromosomal instability [41]. It has been shown that excessive telomere loss is associated with various sperm parameters viz. decreased concentration, low motility and vitality [42,43,44,45], and male infertility [46,47,48,49,50,51,52,53,54,55,56].

Keeping in mind the limited antioxidant activity in seminal plasma, we aimed to conduct the present study to (i) understand the roles of SIRT1 and SIRT3 proteins in regulating levels of DNA damage and telomere length in infertile and fertile men, (ii) determine whether leukocyte telomere length (LTL) is correlated with sperm telomere length (STL) and its prognostic potential for male infertility, and (iii) investigate whether oxidative biomarkers are correlated with semen parameters in infertile men.

## 2. Results

### 2.1. Comparison of Age, Body Mass Index (BMI), Semen Parameters and Other Blood Biomarkers between Oligospermic Men and Fertile Men

The results of the various parameters including age, BMI, paternal age at conception (PAC) and routine semen characteristics are summarized in Table 1. Sperm concentration (31.46 ± 15.63 vs. 181.33 ± 29.01; *p* < 0.009), sperm motility (30.55% ± 5.16 vs. 71.13% ± 3.68; *p* < 0.001) and sperm count (106.1 ± 42.11 vs. 416.46 ± 61.51; *p* < 0.003) were remarkably significantly lower in oligospermic men compared to fertile individuals. In addition, abnormal sperm morphology (98.61% ± 0.21 vs. 96.18% ± 0.14; *p* = 0.002) was significantly higher in the oligospermic group compared to fertile men. Seminal plasma cortisol levels (1.98 µg/dL ± 0.21 vs. 1.51 ± 0.19; *p* = 0.03) and blood homocysteine levels (7.84 µmol/L ± 0.41 vs. 6.32 ± 0.30; *p* = 0.03) were also significantly higher in oligospermic men compared to fertile controls. The data of the main outcome measures (Mean ± SE along with 95% CI) are shown in Table 2.

### 2.2. Assessment of Sperm and Leukocyte Telomere Lengths

We assessed the absolute telomere length in sperms and leukocytes from both groups. STL and LTL were significantly shorter (8154 ± 138.57 bp vs. 13,583 ± 157.1 bp; *p* < 0.0001 and 6612 ± 104.1 bp vs. 7367 ± 122.6 bp vs.; *p* = 0.004, respectively) in oligospermic individuals compared to the fertile group (Figure 1A).

### 2.3. SIRT1 and SIRT3 Protein Levels in Seminal Plasma

The level of SIRT1 protein in seminal plasma was significantly low (5.75 ± 0.15 ng/mL vs. 6.76 ± 0.18 ng/mL; *p* = 0.0003; Figure 1B) in oligospermic patients compared to fertile men. Similarly, the amount of SIRT3 protein in oligospermic men was significantly low (3.92 ± 0.16 ng/mL vs. 4.80 ± 0.12 ng/mL, *p* = 0.001; Figure 1B).

### 2.4. Level of Antioxidants and Oxidative Stress Biomarkers in Seminal Plasma

Oligospermic men had significantly lower levels of TAC (1.12 mM ± 0.04 vs. 2.11 ± 0.04; *p* = 0.0001; Figure 1C), catalase (8.71 U/mL ± 0.34 vs. 20.94 ± 0.41; *p* < 0.0001; Figure 1C) and SOD (11.77 U/mL ± 0.43 vs. 22.24 ± 0.61; *p* < 0.001; Figure 1D), and significantly higher levels of MDA (1.89 µmol/L ± 0.05 vs. 0.92 µmol/L ± 0.04; *p* < 0.001; Figure 1D), in seminal plasma compared to the fertile men.

### 2.5. Assessment of Protamine Deficiency and DNA Fragmentation (DFI) in Sperms

Figure 1E shows the percentages of protamine-deficient sperms (53.27 ± 0.54 vs. 26.44 ± 0.46; *p* < 0.0001) and sperm DFI (% DFI; 12.28 ± 0.48 vs. 6.65 ± 0.16; *p* < 0.0001); both risk factors for fertility were significantly higher in oligospermic men compared to the fertile group.

### 2.6. Correlation between Sperm Parameters, STL, LTL, Sperm Lipid Peroxidation and Protamine Deficiency

In the present study, we observed a significant positive correlation between absolute sperm telomere length and absolute leucocyte telomere length (r = 0.49; *p* = 0.001), and a significant negative correlation between STL and %DFI (r = −0.38; *p* = 0.04) and protamine deficiency (r = −0.50; *p* = 0.001 Table 3). Similarly, LTL also showed a significant negative correlation with %DFI (r = −0.51; *p* = 0.001) and protamine deficiency (r = −0.41; *p* = 0.01). A positive correlation was observed between telomere length and sperm motility (STL; r = 0.26 and LTL; r = 0.198, respectively). Sperm morphology was positively correlated with TL; however, it did not achieve significant levels. We also found a significant positive correlation of TL with sperm concentration (STL; r = 0.56, *p* = 0.001; LTL; r = 0.384; *p* = 0.01) and sperm count (STL; r = 0.43; *p* = 0.01; LTL; r = 0.49; *p* = 0.01, respectively). The percentage of DNA fragmentation (%DFI) showed negative correlations with sperm concentration and sperm count. However, protamine deficiency (%) showed a significant negative correlation with sperm concentration and count (r = −0.60; *p* < 0.001 and r = −0.48; *p* < 0.001, respectively). A strong significant positive correlation was observed between %DFI and protamine deficiency (r = 0.63; *p* < 0.001).

### 2.7. Correlation between Seminal Plasma Antioxidant Levels, Oxidative Stress Biomarkers, Lipid Peroxidation and Sperm Parameters

A positive correlation was observed for TAC, SOD and CAT levels with sperm motility (Table 4 ). Lipid peroxidation was strongly but negatively correlated with sperm motility (r = −0.46; *p* = 0.01). TAC showed a significant negative correlation with abnormal sperm morphology (r = −0.44; *p* = 0.01). Sperm concentration also showed a significant positive correlation with TAC (r = 0.35; *p* = 0.04), SOD (r = 0.56; *p* < 0.001) and CAT (r = 0.38; *p* = 0.01), whereas lipid peroxidation (i.e., MDA) was significantly negatively correlated with sperm concentration (r = −0.60; *p* < 0.001). DNA fragmentation also showed a significant negative correlation with TAC, SOD and CAT. However, DNA fragmentation showed a strong positive correlation with lipid oxidation (i.e., MDA; r = 0.50; *p* < 0.001; Table 4).

### 2.8. Correlation of SIRT1 and SIRT3 Proteins with Sperm Parameters, DFI and STL

Both SIRT1 and SIRT3 showed significant negative correlations with abnormal sperm morphology (r = −0.38; *p* = 0.01 and r = −0.37; *p* = 0.01, respectively, Table 5 ). Sperm count and motility were found to have positive correlations with SIRT1 (r = 0.38; *p* = 0.01; r = 0.47; *p* = 0.001, respectively) and SIRT3 (r = 0.40; *p* = 0.01; r = 0.42; *p* = 0.001, respectively, Table 5 ). Both SIRT1 and SIRT3 showed significant but negative correlations with %DFI (r = −0.59 and r = −0.62, respectively, Table 5). These also show similar associations with oxidative stress biomarkers. SIRT1 and SIRT3 are positively correlated with STL (r = 0.39; *p* = 0.01; r = 0.21, respectively, Table 5). These findings indicate a moderate or strong association of SIRT1 and SIRT3 with all sperm parameters we investigated.

## 3. Discussion

In the last decade, the role of telomere biology has gained much attention in the field of human fertility [52,57,58,59]. Telomere shortening and/or its dysfunction is a useful biomarker of aging, cancer, age-related chronic disorders and reproduction fitness. Telomeres play a vital role in maintaining genomic integrity and chromosomal stability, and ensure proper chromosome alignment during DNA replication [60,61,62]. Shortened telomeres might impede spermatogenesis by inducing segregation errors during mitosis, cell cycle delays, reduced regenerative capacity and cell death of germ cells, and as a result, adversely affect various semen parameters including sperm count [42,52]. In this study, we demonstrated the shortening of telomeres in the sperm of oligospermic men, which may be caused by lower levels of antioxidants as well as low protein levels of SIRT1 and SIRT3 in seminal plasma. In addition, we have also shown that SIRT1 and SIRT3 protein levels are strong indicators of sperm quality.

Although telomere shortening is an inexorable process of cellular aging, genetic background, diet, stress (psychological as well as oxidative), lifestyle and environmental factors may also influence TL shortening [58,63,64,65,66,67].

Many studies suggest that telomere length is significantly associated with sperm count, concentration, motility, and morphology [43,44,45,52,53,68,69]. We found that STL is significantly shorter in infertility patients and is strongly inversely correlated with various semen parameters. It is also positively correlated with LTL. It has been shown that sperm telomeres are compacted and packaged onto small basic proteins known as protamine [70,71]. Many studies have shown that sperm protamine deficiency is strongly associated with STL [44,50,52,53]. The results obtained from the present study show a strong correlation of protamine deficiency with STL. Chromatin components of mature sperm are rich in protamines, and the latter help in stabilizing sperm chromatin [72,73,74]. It has been shown that sperm nuclei with inadequate amounts of protamine are prone to chemical disruption [75], and protamine deficiency also induces marked changes in chromatin packing, as well as in the shape and structure of the acrosome [74,76]. Therefore, it may be possible that sperms with shorter TL have higher amounts of abnormal chromatin packing compared to those with normal/longer TL, which could expose DNA to genotoxic stresses. This statement is corroborated by the findings of significant correlations between sperm DNA damage and protamine deficiency [44,50,52,53]. It has been shown that telomeres play an important role in chromosome reorganization via their attachment to the nuclear envelope (NE) during meiosis. Telomeres cluster in proximity to the centrosome during meiotic leptotene, forming the telomere bouquet [77]. Even though this bouquet of telomeres disseminates after the zygotene stage of meiosis, telomere-mediated connections with the SUN/KASH domain proteins in the NE remain throughout meiosis, connecting chromosomes with the cytoskeleton to promote chromosome motion [78]. Chromatin organization in the sperm varies significantly compared to somatic cells, as DNA is tightly packed with protamines in sperm, but in somatic cells DNA is packaged less tightly on histones depending on the methylation and acetylation levels of histones [79,80]. In spite of this, the territorial organization and non-random intra-nuclear arrangement of intra-nuclear positioning of chromosomes (CHRs) are preserved [81,82,83]. This process then allows a crossing over between these chromosomes, thereby securing chromosome segregation during the first meiotic division to prevent lethal chromosomal anomalies [84]. Therefore, accelerated telomere shortening may end up causing poor gamete viability, poor pairing level, delayed pairing time, incomplete synapsis, delayed synapsis formation and reduced recombination [85]. Dysfunctional telomeres have the potential to affect sperm fertilization and chromosome pairing, thus resulting in recombination defects, aneuploidy and low sperm count [86,87,88].

Our results show that infertile men had a significantly higher sperm DNA fragmentation index than fertile men, which is consistent with the findings of previous studies [44,56,89,90]. It has been shown that oxidative stress induces significant damage to chromatin integrity and STL due to the generation of reactive oxygen species (ROS) [43,44,50,52]. ROS are generated from white blood cells and immature sperms with abnormal morphologies due to activation by inflammatory processes and toxic exposure in the male uro-genital tract [91,92]. Moreover, sperm have an extremely reduced cytoplasm, and as a result have low amounts of antioxidants [91,93]. It is proposed that sperm DNA fragmentation can occur due to unrepaired meiotic breaks during the meiotic process, the inefficient removal of apoptotic germ cells and necrotic cells during the epididymal maturation process, and the replacement of histones by protamines during spermatogenesis [94]. It has been shown recently that ROS induces abnormal sperm function, motility and morphology due to the inadequate antioxidant capacity of seminal plasma [95]. The results obtained from our study clearly demonstrate that %DFI, protamine deficiency and MDA (lipid peroxidation) levels are significantly higher, whereas TAC, CAT and SOD are significantly lower, in infertile men compared to fertile men. Our results are in agreement with previous published reports on SOD [50,96,97,98] and CAT [96,97,98,99,100]. These findings support the notion of the protection of sperm against perceived oxidative stress by the antioxidant enzymes during spermatogenesis, especially due to the inability of the mature sperm to synthesize these enzymes [101]. Our results show a strong correlation between protamine deficiency, DNA fragmentation and MDA (lipid peroxidation), and are consistent with published reports, thereby confirming that low levels of protamine and sperm immaturity leave them susceptible to lipid peroxidation and DNA damage because of chromatin loosening, which allows for the easier access of genotoxicants to interact adversely with DNA [102,103].

SIRT1 and SIRT3 proteins are found in seminal plasma, and their levels in seminal plasma are inversely correlated with oxidative stress biomarkers and the amount of sperm DNA fragmentation, as evident in our study. These findings are in agreement with the previously published reports [31,32,104,105]. Seminal plasma provides nutrients for the maintenance of the maturing sperms, and the increased generation of ROS could result in reduced sperm metabolism and diminish motility, thus leading to infertility [106,107]. It has been shown that the expression and activation of SIRT1 is influenced by the generation of ROS [108,109,110]. SIRT1 and SIRT3 may play not only an important role in spermatid differentiation during the spermatogenesis process, but also alleviate oxidative stress levels, and their absence increases susceptibility to DNA damage [111]. For proper sperm motility, sirtuins must be expressed in testes [15]. It has been shown that SIRT1 knock-out mice (*Sirt*^−/−^) have abnormal seminiferous tubules with decreased sperm counts, thus suggesting that SIRT1 deficiency leads to defective spermatogenesis, including spermatogenesis arrest during the late meiotic prophase [15,112]. These animals also show significantly higher numbers of sperms with single- or double-strand DNA breaks [112]. In addition, these knock-out mice also show increased numbers of apoptotic cells in the seminiferous tubules [113]. Therefore, it is plausible that a deficiency of sirtuins including SIRT 1 [114] may either directly or indirectly induce higher oxidative stress leading to sperm DNA damage, or alternatively may make sperm DNA, or telomeric DNA specifically, less susceptible to genotoxic stress by increasing the compaction of heterochromatin via the deacetylation of histones [115,116] (Figure 2). This is plausible because the telomeric DNA in sperm is associated with histones, not protamines [117,118].

Our results, in combination with previous reports, demonstrate that the correlation between sirtuins and oxidative stress or telomeric shortening is inverse [109]. Therefore, it is likely that both the quantity and activity of sirtuins can be impacted by increased oxidative stress via influencing their expression and modifying protein interaction post-transcriptionally [109]. These observations also lead to the alternative hypothesis that low levels of SIRT1 and SIRT3 can affect the expression and activation of antioxidant proteins such SOD and CAT [109,119]. Furthermore, SIRT1 1 has been shown to prevent telomere attrition by regulating the transcription and translation of hTERT, and regulating telomere-protective proteins and DNA repair proteins [34,120,121]. SIRT1 may also exert these effects by deacetylating telomere-associated histones, causing the compaction of telomeric chromatin. Oxidative stress is the result of an imbalance between the rate of production of reactive oxygen and nitrogen species during cell metabolism and the state of the antioxidant machinery within the cell, leading to oxidative damage to a wide range of biomolecules, including DNA [122]. Oxidized DNA bases inhibit telomerase activity and thus reduce the rate of telomere lengthening, causing DNA replication stress and leading to DNA breaks in telomeres [123,124]. A negative link between oxidative stress and TL was supported by experiments wherein pro-oxidative agents induced higher oxidative damage in telomeres than other chromosome regions [125]. A lower activity of intracellular antioxidant enzymes such as superoxide dismutase has been linked to telomere shortening [126]. It has also been shown that antioxidants such as vitamins C and E, and glutathione (main intracellular antioxidants), can reduce telomere attrition [127,128,129].

Overall, the findings obtained in the present study provide further evidence that higher levels of oxidative stress, protamine deficiency, and low levels of antioxidants in seminal plasma have the potential to damage sperms and accelerate telomere attrition. In addition, lower levels of sirtuins such as SIRT1 and SIRT3 in seminal plasma or within sperm can exacerbate damage to sperm DNA, thus making it unfit for proper fertilization, leading to male infertility. The results of our study indicate that (i) telomere shortening is likely to be a fundamental causative factor of sperm dysfunction and insufficiency; (ii) it is plausible that excessive telomere shortening may be due to increased oxidative damage to DNA, and (iii) deficiencies in SIRT 1 and SIRT 3 may contribute substantially to excessive oxidation by impairing mitochondrial function efficiency. The clinical implication of these observations is that the vicious sequence of SIRT 1 and SIRT3 deficiency, oxidative stress and telomere shortening needs to be broken in infertile patients if their reproductive capacity is to be regenerated successfully. Further improving this knowledge is critical for identifying which nutritional factors are essential to restoring spermatogenesis successfully.

## 4. Materials and Methods

### 4.1. Study Participants

The study population consists of 272 subjects diagnosed with oligospermia and 251 fertile men who have fathered at least one child. Blood and semen samples were obtained from the study participants following approval from the Human Research Ethics Committee. Informed consent was obtained from each subject involved in the study. All the included subjects had a history of infertility for more than 12 months (i.e., were unable to conceive at least once after 12 months or more of regular unprotected sexual intercourse) and had not received exogenous hormonal drugs, chemotherapy, or other medicines known for impairing testicular functions within the past 6 months before semen collection. Fertile men (control group) had fathered at least one child and had normal sperm parameters. Infertility patients who had been diagnosed with conditions such as cryptorchidism, varicocele, endocrine disorders, Klinefelter’s syndrome, testicular size, anatomical disorders, azoospermia, previous history of scrotal trauma or surgery, and those carrying Y-chromosome microdeletions were excluded from the study. In addition, patients with a history of urogenital infection, orchitis, erectile dysfunction, alcohol consumption, testicular pathology, sexually transmitted disease, drug abuse, vasectomy, chronic diseases, exposure to hazardous chemicals such as lead, arsenic and pesticides, etc., and radiation therapy were not included in the present study. Data for life style factors such as diet, stress, smoking, physical activity and alcohol consumption were collected and analyzed. We tried our best to match these numbers as closely as possible when recruiting the control group (fertile men). Analyses of semen and various biochemical parameters from the blood were performed as per standard methods. Sperm parameters such as sperm concentration, sperm motility and morphology were classified as per World Health Organization (WHO, 2010) guidelines [130]. Semen samples were obtained by masturbation after 2–5 days of abstinence. Overnight fasting blood samples were also collected from each participant. Biochemical parameters were analyzed by a certified diagnostic laboratory.

### 4.2. Semen Analysis

Semen analysis (concentration, motility and morphology) was carried out as per standard guidelines as mentioned in the WHO protocol at the certified diagnostic laboratory. Semen samples were also subjected to other assays: STL, SIRT1 and SIRT3 protein quantification, antioxidant analysis (TAC, CAT and SOD), lipid peroxidation assay (MDA), protamine deficiency and sperm DNA fragmentation index (%DFI).

### 4.3. Sperm Telomere and Leukocyte Telomere Length Measurements

DNA was extracted from sperms (20 × 10^6^) and blood using Qiagen Blood and Tissue Midi kit (Qiagen, Germantown, MD USA) as per the manufacturer’s instructions. Sperm telomere length (STL) and leucocyte telomere length (LTL) were determined by real-time polymerase chain reaction (qRT-PCR), as described previously [131,132,133]. Semen was collected by masturbation. Sperm samples were incubated overnight at 4 °C in lysis buffer containing RNAse (10 mg/mL) and proteinase K (1 mg/mL). Dithiothreitol (5 mM; Sigma-Aldrich, Sydney, Australia) was also added to the solution to facilitate the complete lysis of the spermatozoa. The contents were incubated at 4 °C for one hour followed by one hour of incubation at 37 °C. DNA was then extracted using a Qiagen kit as per manufacturer’s instructions. Blood was collected in EDTA tubes to isolate DNA and genomic DNA was extracted from isolated leucocytes using the QIAamp DNA blood mini kit (Qiagen, Germantown, MD, USA). Purified DNA samples were quantified using a NanoDrop 1000 spectrophotometer (Thermo Fisher Scientific, Waltham, MA, USA) and diluted as per experimental requirements (5 ng/μL). Telomere length was measured using quantitative real-time PCR as described previously [131]. The ratio of the telomere (T) repeat copy number to the single-copy gene (S) was determined for each sample using an ABI 7300 Real-Time PCR Detection System (Life Technologies, Waltham, MA, USA). The final concentrations of the PCR reagents were 1 × SYBR Green Mix (Life Technologies, Waltham, MA, USA), 20 ng DNA, 0.2 μmol of telomere specific primers (F: 5′-GGTTTTTGAGGGTGAGGGTGAGGGTGAGGGTAGGGT-3′; R: 5′-TCCCGACT ATCCCTATCCCTATCCCTATCCCTATCCCTA-3′) and 0.3 μmol of 36B4 primers (F: 5′-CAGCAAGTGGGAAGGTGTAATCC-3′; R: 5′-CCCATTCTATCATCAACGGGTC AA-3′). The reactions were performed using telomere and 36B4 specific primers in a 96-well plate, and each plate included a reference DNA sample. It may be noted that primers 36B4 can amplify pseudogene regions on chromosomes 1, 2, 5, 12 and 18. A five-point serial dilution standard curve of DNA concentration versus T/S ratio using DNA isolated from the 1301 cell line (which has a mean telomere length of 23,000 base pairs) was established in each plate. The standard curve was then used to convert the T/S ratio into telomere length (TL) in base pairs (bp) using the following equation: Absolute TL (bp) = 2433.23X + 3109.51, where X = T/S ratio, 2433.23 is the slope and 3109.51 is the intercept of the standard curve. To generate the standard curve and convert T/S ratio into absolute TL (bp), DNA isolated from the IMR90 cell line was used. DNA was isolated from cells taken from the cultured cell line at different population doubling times (~35 h; 10 time points). The Southern blot method (TRF) was used to determine the TL of these DNAs isolated from 10 different time points of sub-culturing. DNA isolated from these cells was also used in the qPCR assay to determine T/S ratios in part to generate the initial standard curve, which was then converted into base pairs. The formula generated from this was later used to convert T/S ratios to absolute telomere length: absolute TL (bp) = 2433.23X + 3109.51 where X = T/S ratio of samples, 2433.23 is the slope and 3109.51 is the intercept [132,133]. A standard curve with a high correlation factor (R^2^ ≥ 0.97) was required to accept the results from the plate. The intra-assay coefficient between triplicates was 2.9% for telomeres and 1.7% for the single-copy gene, whereas the inter-assay CV between plates was 0.6% for telomeres and 0.73% for the single-copy gene.

### 4.4. SIRT1 and SIRT3 Protein Quantification in Seminal Plasma

Quantities of SIRT1 and SIRT3 proteins in the seminal plasma samples were measured by enzyme-linked immune sorbent assay as per manufacturer’s instructions (ELISA) kit (SIRT1 Human Simp, eStep ELISA^®^ Kit; ab171573, Abcam, MA, USA; SIRT3/Sirtuin 3 ELISA kit, LifeSpan BioSciences, Seattle, WA, USA, respectively).

### 4.5. Measurement of Antioxidant and Oxidative Stress Biomarkers

#### 4.5.1. Measurement of Total Antioxidant Capacity (TAC)

TAC was measured using the Ferric Reducing Antioxidant Power (FRAP) assay [134]. This protocol measures the change in absorbance at 593 nm due to the formation of Fe++ tripyridyltriazine (blue color) from the Fe+++ tripyridyltria-zine (colorless) form due to the action of electron donating antioxidants. Briefly, 50 µL of seminal plasma was mixed with 1 mL of FRAP reagent and the absorbance was measured at 593 nm.

#### 4.5.2. Measurement of Catalase Activity

Catalase (CAT) activity in seminal plasma was measured as per the previously published method [135] by monitoring the initial rate of reduction of hydrogen peroxide at 240 nm wavelength expressed as Enz. Ut/L.

#### 4.5.3. Measurement of SOD Activity

SOD activity in human seminal plasma was measured using a kit supplied by Randox Laboratories, Crumlin, UK, with xanthine and xanthine oxidase to generate superoxide radicals that react with 2-(4-ido-phenyl)-3-(4-nitrophenol)-5-phenyl-2h-tetrazolium chloride (INT) to form a red formazan dye. Seminal plasma samples were diluted 3-fold using l0 mM phosphate buffer (pH 7.0). One unit of SOD is the amount that induced a 50% inhibition in INT reduction rate.

#### 4.5.4. Measurement of Malondialdehyde Levels

The lipid peroxidation level in seminal plasma was measured according to the levels of MDA [136]. Briefly, reverse-phase high-performance liquid chromatography (HPLC; Agilent Technologies 1200 series) with a fluorescence detector using EC 250/4.6 Nucleodur^®^100-5 C18ec column (Macherey-Nagel, Duren, Germany) (mobile phase 60:40 *v*/*v* of methanol:buffer (50 mM potassium monobasicphosphate buffer, pH 6.8); detection at Ex515-Em555) was used to measure MDA levels.

### 4.6. Sperm Protamine Deficiency Assay

A sperm protamine deficiency assay was performed as per the previously published method [137]. Semen samples were first fixed in a carnoy’s solution (fixative consisting of methanol/glacial acetic acid 3:1; ice cold solution for 5 min). After two washes with PBS, the slides were stained with chromomycin A3 (CMA3). For CMA3 (Sigma) staining, each slide was treated for 15 min with 150 μL of CMA3 solution (0.25 mg/mL in McIlvain buffer: 7 mL citric acid (0.1 M), 32.9 mL Na_2_HPO_4_·7H_2_O (0.2 M), pH 7.0 and 10 mM MgCl_2_)). The slides were rinsed again with buffer and mounted with buffered glycerol (1:1). The scoring of the slides was performed on a fluorescent Zeiss axioplane microscope (Oberkochen, Germany; 460–470 nm filters) and at least 500 sperms were counted. Sperms with a light yellow color are here referred to as protamine deficient and a percentage of these sperms is calculated.

### 4.7. Sperm DNA Fragmentation (SDF) Assay

The sperm DNA fragmentation assay was performed using the terminal deoxynucleotidyl transferase–mediated deoxyuridine triphosphate nick-end labeling (TUNEL) assay as per the previously published method [138]. The semen sample was first incubated at 37 °C for half an hour with LIVE/DEAD^®^ Fixable Dead Cell stain (ThermoFisher Scientific, Waltham, MA, USA). The cells were washed twice with PBS. The cells (2 × 10^6^) were treated first with 2 mM dithiothreitol (DTT, Sigma-Aldrich, St. Louis, MO, USA) for 45 min, washed twice with PBS and fixed in 3.7% formaldehyde solution (Sigma-Aldrich, St.Louis, MO, USA) followed by permeabilization treatment (100 mg Sodium citrate, 100 µL Triton X-100 in 100 mL MilliQ water; incubated at 4 °C for 5 min). Sperm were then washed twice with PBS and analyzed using the fluorescein In Situ Cell Death Detection Kit (Roche Diagnostics, Mannheim, Germany) using Accuri C6 flow cytometer (BD BioSciences, Franklin Lakes, NJ, USA). Sperm DNA fragmentation was analyzed in the total sperm sample at a flow rate of 35 µL/min, recording 5000–10,000 events/sample. The positive control was treated with DNase I (Qiagen, Germantown, MD, USA), whereas the negative control (all components except the terminal deoxynucleotidyl transferase enzyme) was included in the analysis. The method has been standardized and threshold values of normality (≤13% SDF) established compared to a fertile cohort as per the previously published method [139]. Total sperm DNA fragmentation (percentage of the sperms that are positive for DNA fragmentation) was calculated for each sample.

### 4.8. Statistical Analysis

Parametric statistical methods were used for various biomarkers that exhibited Gaussian distribution. Correlation analysis was performed by Pearson’s test. Descriptive statistics were used to summarize demographic characteristics. The unpaired t-test was used to study the differences in various biomarkers in both groups. Groups (infertile and fertile) were matched with respect to age and various lifestyle factors such as smoking, stress, physical activity and alcohol consumption. Data have been expressed as standard error of the mean (means ± SEM)

Pearson correlation was carried out to analyze the association between different parameters. Statistical analyses were performed using GraphPad Prism (Version 9.0; San Diego, CA, USA) and SPSS (IBM SPSS Version 23; Sydney, Australia). Significance for all statistical tests was set at *p* < 0.05 for all analyses.

## 5. Conclusions

We found a positive correlation of both SIRT1 and SIRT3 with sperm motility and concentration, and a negative correlation with abnormal sperm morphology. In addition, we also found a negative correlation between sirtuins and DNA fragmentation. Hence, it can be concluded that lower seminal plasma levels of SIRT1, SIRT3, TAC, SOD and catalase may result in increases in oxidative stress, which impacts on various sperm parameters, thereby leading to male infertility (Figure 3). It has been shown previously that SIRT1 increases mitochondrial metabolism, which results in lowering oxidative stress [140]. Therefore, it is important to consume a nutritious diet rich in antioxidants and sirtuin activators (that include polyphenol compounds such as resveratrol and sirtuin co-factors such as NAD+) and lead an active life; a healthy life-style will also help in alleviating oxidative stress by increasing the levels of sirtuins and/or their activity to reduce the telomere attrition rate and reduce the risk of male infertility. Sirtuins helps in conserving genomic stability and play an important role in chromatin compaction. Furthermore, to accomplish efficient reproduction, genomic stability should be maintained by protecting the spermatogenic germ cells from DNA damage [141].

## Figures and Tables

**Figure 1 ijms-25-00718-f001:**
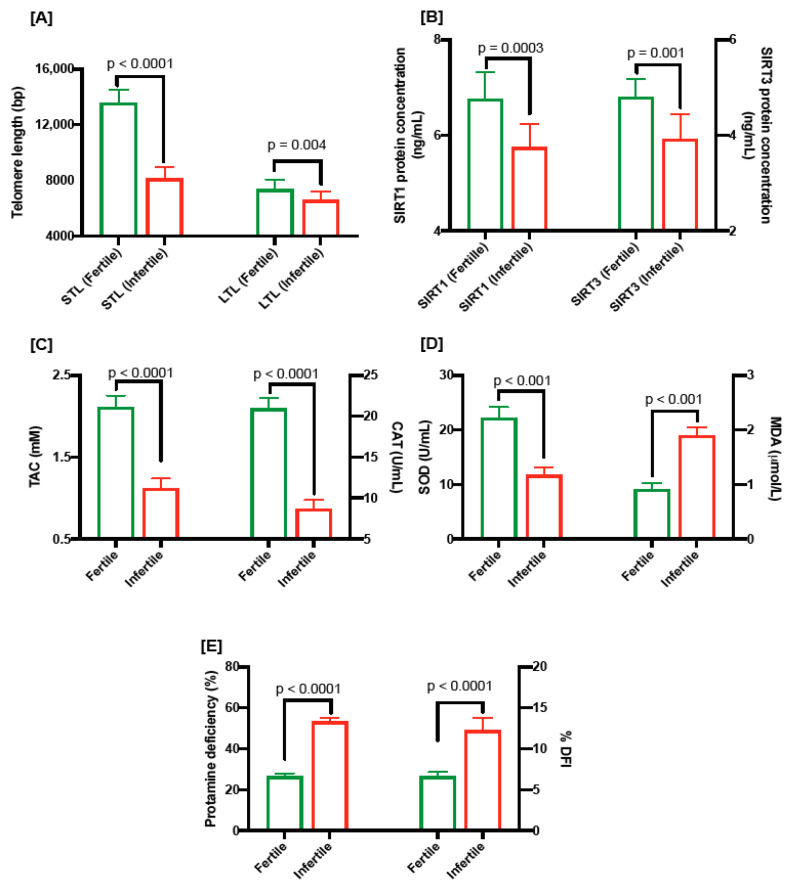
(**A**) Comparison of sperm telomere length and leukocyte telomere length in fertile and infertile men. (**B**) SIRT1 (ng/mL) and SIRT 3 (ng/mL) levels in fertile and infertile men. (**C**) TAC levels (mM; left) and CAT levels (U/mL) in fertile and infertile men. (**D**) SOD levels (U/mL; left) and MDA levels (µmol/L) in fertile and infertile men. (**E**) protamine deficiency (%; left) and DNA damage as shown by DNA fragmentation index (%DFI; right) in fertile and infertile individuals.

**Figure 2 ijms-25-00718-f002:**
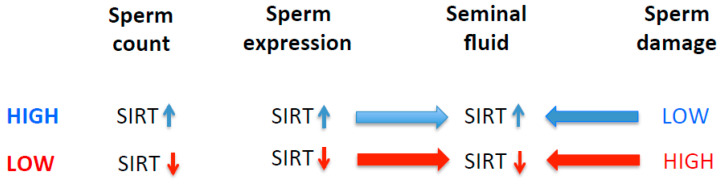
Proposed association of SIRT with sperm parameters such as sperm count and DNA damage.

**Figure 3 ijms-25-00718-f003:**
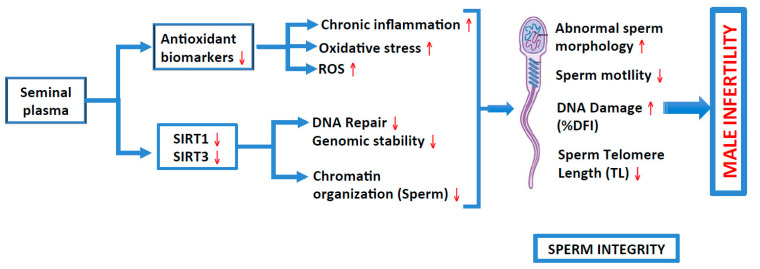
Roles of antioxidant biomarkers and sirtuins (1 and 3) in inducing inflammation, oxidative stress, generation of ROS and DNA repair, and of genomic instability and chromatin organization, respectively, on sperm integrity (various sperm parameters), thus causing male infertility.

**Table 1 ijms-25-00718-t001:** Clinical characteristics of the study participants.

Parameters	Fertile Group (n = 251)	Infertile Group (n = 272)	*p*-Value
Age (in years)	31.52 ± 4.35	32.72 ± 4.91	0.68
BMI (kg/m^2^)	24.03 ± 2.91	25.12 ± 2.45	0.73
Paternal age at conception (years)	29.14 ± 5.62	27.81 ± 4.94	0.25
Duration of marriage (years)	6.18 ± 2.72	7.01 ± 1.98	0.09
Total testosterone (ng/mL)	4.38 ± 0.12	4.58 ± 0.14	0.12
Triglyceride (mg/dL)	129.5 ± 7.16	149.43 ± 6.28	**0.03**
Cholesterol (mg/dL)	191.4 ± 15.16	193.6 ± 19.18	0.84
HDL (mg/dL)	47.16 ± 5.18	46.38 ± 6.05	0.79
LDL (mg/dL)	123.4 ± 11.24	129.65 ± 18.08	0.91
Semen volume (mL)	3.34 ± 0.31	3.14 ± 0.28	0.18
Sperm concentration (10^6^/mL)	181.33 ± 29.01	31.46 ± 15.63	**0.009**
Semen pH	7.8 ± 0.2	8.01 ± 0.3	0.38
Sperm count (10^6^/ejaculation)	416.46 ± 61.51	106.1 ± 42.11	**0.003**
Sperm motility (%)	71.13 ± 3.68	30.55 ± 5.16	**0.001**
Abnormal sperm morphology (%)	96.01 ± 0.14	98.68 ± 0.21	**0.002**
Cortisol in seminal plasma (µg/dL)	1.51 ± 0.19	1.98 ± 0.31	**0.01**
FSH (mIU/mL)	4.15 ± 0.13	4.85 ± 0.19	**0.043**
LH (IU/L)	4.09 ± 0.09	4.32 ± 0.12	0.38
Vitamin D in seminal plasma (ng/mL)	27.83 ± 8.16	28.19 ± 10.06	0.81
Vitamin B12 (pmol/L)	419.2 ± 15.35	403.5 ± 12.15	0.39
Homocysteine (µmol/L)	6.32 ± 0.3	7.84 ± 0.41	**0.03**
Serum folate (nmol/L)	32.1 ± 0.4	31.8 ± 0.5	0.89

**Table 2 ijms-25-00718-t002:** Main outcome measures in two groups (mean ± SE along with 95% CI values).

Parameters	Fertile Group (n = 251) Mean ± SE (95% CI)	Infertile Men (n = 272) Mean ± SE (95% CI)
Telomere length (bp)	13,583 ± 157.1 bp (13,274–13,912)	8154 ± 138.57 (7867–8435)
SIRT1 (ng/mL)	6.76 ± 0.55 (6.36–7.156)	5.75 ± 0.48 (5.407–6.093)
SIRT3 (ng/mL)	4.80 ± 0.39 (4.524–5.076)	3.92 ± 0.51 (3.55–4.289)
TAC (mM)	2.11 ± 0.04 (2.015–2.211)	1.12 ± 0.04 (1.033–1.202)
Catalase (U/mL)	20.94 ± 0.41 (20.02–21.86)	8.71 ± 0.34 (7.937–9.485)
SOD (U/mL)	22.24 ± 0.62 (20.85–23.62)	11.77 ± 0.43 (10.80–12.73)
MDA (µmol/L)	0.92 ± 0.04 (0.835–0.995)	1.89 ± 0.05 (1.791–2.003)
Protamine (%)	26.44 ± 0.46 (25.39–27.49)	53.27 ± 0.54 (52.05–54.50)
DFI (%)	6.65 ± 0.16 (6.282–7.020)	12.28 ± 0.48 (11.18–13.34)

**Table 3 ijms-25-00718-t003:** Correlation between sperm telomere length (STL), lymphocyte telomere length (LTL), DNA fragmentation, protamine deficiency, and sperm parameters.

Parameters	STL	LTL	DNA Fragmentation	Protamine Deficiency
STL	-	0.49 ***	−0.38 *	−0.49 ***
LTL	0.49 ***	-	−0.51 ***	−0.41 **
Sperm motility (%)	0.26	0.20	−0.32	−0.46 **
Abnormal sperm morphology (%)	−0.27	−0.29	0.48 ***	0.33
Sperm concentration (10^6^/mL)	0.56 ***	0.38 **	−0.31	−0.60 ***
Sperm count (10^6^/ejaculation)	0.43 **	0.49 **	−0.34	−0.48 ***
Protamine deficiency (%)	−0.49 ***	−0.41 **	0.63 ***	-
DNA fragmentation (%)	−0.38 *	−0.51 ***	-	0.63 ***

* *p* < 0.05; ** *p* < 0.01; *** *p* < 0.001.

**Table 4 ijms-25-00718-t004:** Correlation between seminal plasma antioxidant, oxidative stress biomarkers and sperm parameters.

Parameters	TAC	SOD	CAT	MDA
Sperm motility (%)	0.31	0.25	0.31	−0.46 **
Abnormal sperm morphology (%)	−0.45 **	−0.27	−0.29	0.31
Sperm concentration (10^6^/mL)	0.35 *	0.56 ***	0.38 **	−0.60 ***
Sperm count (10^6^/ejaculation)	0.38 *	0.41 **	0.47 **	−0.49 ***
DNA fragmentation (%)	−0.41 **	−0.38 *	−0.51 ***	0.50 ***

* *p* < 0.05; ** *p* < 0.01; *** *p* < 0.001.

**Table 5 ijms-25-00718-t005:** Correlation of SIRT1 and SIRT3 proteins with sperm parameters.

Parameters	Abnormal Sperm Morphology (%)	Sperm Motility (%)	Sperm Concentration (10^6^/mL)	Sperm Count (10^6^/ejaculate)	DFI (%)	STL
SIRT1	−0.38 **	0.47 **	0.29	0.38 **	−0.59 ***	0.39 **
SIRT3	−0.37 **	0.42 **	0.31	0.40 **	−0.62 ***	0.21

** *p* < 0.01; *** *p* < 0.001.

## Data Availability

Data will be uploaded to a publicly available repository upon acceptance of the manuscript.

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
