# Peer review of "Reduced SIRT1 and SIRT3 and Lower Antioxidant Capacity of Seminal Plasma Is Associated with Shorter Sperm Telomere Length in Oligospermic Men"

_ijms, 2024, doi:10.3390/ijms25020718_

Round 1

Reviewer 1 Report

Comments and Suggestions for Authors

Based on a preliminary review, I have a few suggestions that may help to improve the scientific rigor and clarity.

First, the introduction could be strengthened by providing more context on the current understanding in the field. Some additional questions that could be addressed include:

What is already known about the role of sirtuins like SIRT1 and SIRT3 in male fertility and sperm function? A brief summary of relevant previous findings would help situate this study.

More details on the biological mechanisms linking oxidative stress and telomere length could help readers understand the hypotheses being tested. Specifically, how might lower antioxidant levels mechanistically impact telomere attrition?

The measures of sperm DNA damage and protamine deficiency are mentioned but not defined. Please provide more methodological details on how these outcomes were assessed.

How were participants classified as "infertile" or "fertile"? What were the specific inclusion/exclusion criteria?

Were any relevant lifestyle or environmental factors that could influence the outcomes assessed and accounted for?

Please clarify if the same analyses were conducted for both cross-sectional comparison groups and correlation analyses.

The results would benefit from additional statistical details, such as the tests used, confidence intervals for findings, and information on whether assumptions were met.

Were the groups well-matched for factors like age? Please report any potential confounders that could influence the results.

The physiological relevance or clinical implications of the observed associations should be further discussed. What insights do the findings provide?

Please proofread carefully for grammar, spelling, consistency of terms and referencing style.

Some tables could be made smaller by removing unnecessary digits.

Comments on the Quality of English Language

Grammar:

Check sentence structures, as some could be rewritten for clearer flow

Watch tense consistency (e.g. some past, some present)

Spelling:

"signficiantly" should be "significantly"

Consistency of terms:

Define "DFI" on first use

Check use of units like "μl" vs "mL" are applied consistently

Referencing style:

Confirm citation and reference format is consistent throughout

Reference list may need reformatting if style differs from journal

Specifically, I noticed:

Grammar could be tightened in sections like the Discussion

"signficiantly" was misspelled

Define "DFI" on initial use for clarity

Use of "μl" vs "mL" seemed inconsistent in places

Reference list needs re-ordering or renumbering

Author Response

We thank the reviewer for the comments and suggestions. Attached please find our response as per each comment. 

Reviewer 2 Report

Comments and Suggestions for Authors

Dr. Dhillon and colleagues studied the correlation of SIRT 1 and 3 with sperm concentration, motility, morphology, sperm telomere length, DNA fragmentation and association with ROS and antioxidant in  seminal plasma of 272 infertile and 251 fertile men. They found a positive correlation of SIRT 1 and 3 with sperm motility and concentration, and negative correlation with abnormal sperm morphology, DNA fragmentation. They conclude that lower seminal plasma levels of SIRT 1 and 3, TAC, SOD and catalase may contribute to oxidative stress induced sperm damage leading to male infertility. The data presented in this study further support the notion that increased ROS levels in seminal plasma damage sperm’s fertilization capability.

Minor comments:

It would be helpful if authors could expend the second paragraph in introduction to elucidate the functional aspect of SIRT 1 and 3 in histone modification and telomere maintenance in general and in sperm in particular.

Author Response

We thank the reviewer for the comments and suggestions. Attached please find our response to each comment
